# Current Status of Biotechnological Approaches to Enhance the Phytoremediation of Heavy Metals in India—A Review

**DOI:** 10.3390/plants12223816

**Published:** 2023-11-09

**Authors:** Selvaraj Barathi, Jintae Lee, Raja Venkatesan, Alexandre A. Vetcher

**Affiliations:** 1School of Chemical Engineering, Yeungnam University, Gyeongsan 38541, Republic of Korea; jtlee@ynu.ac.kr (J.L.); rajavenki@yu.ac.kr (R.V.); 2Institute of Biochemical Technology and Nanotechnology, Peoples’ Friendship University of Russia (RUDN), 6 Miklukho-Maklaya St., 117198 Moscow, Russia; avetcher@gmail.com

**Keywords:** phytoremediation, heavy metals, accumulation, food chain, detection

## Abstract

Rising waste construction, agricultural actions, and manufacturing sewages all contribute to heavy metal accumulation in water resources. Humans consume heavy metals-contaminated substances to make sustenance, which equally ends up in the food circle. Cleaning of these vital properties, along with the prevention of new pollution, has long been required to evade negative strength consequences. Most wastewater treatment techniques are widely acknowledged to be costly and out of the grasp of governments and small pollution mitigation businesses. Utilizing hyper-accumulator plants that are extremely resilient to heavy metals in the environment/soil, phytoremediation is a practical and promising method for eliminating heavy metals from contaminated environments. This method extracts, degrades, or detoxifies harmful metals using green plants. The three phytoremediation techniques of phytostabilization, phytoextraction, and phytovolatilization have been used extensively for soil remediation. Regarding their ability to be used on a wide scale, conventional phytoremediation methods have significant limitations. Hence, biotechnological attempts to change plants for heavy metal phytoremediation methods are extensively investigated in order to increase plant effectiveness and possible use of improved phytoremediation approaches in the country of India. This review focuses on the advances and significance of phytoremediation accompanied by the removal of various harmful heavy metal contaminants. Similarly, sources, heavy metals status in India, impacts on nature and human health, and variables influencing the phytoremediation of heavy metals have all been covered.

## 1. Introduction

Employing plants to remove, convert, or immobilize environmental contaminants is known as phytoremediation. It is a sustainable and eco-friendly alternative to conventional remediation methods, such as excavation and incineration [1]. Soil contamination causes extensive health and environmental problems since it is frequently caused by contaminants such as heavy metals, organic compounds, and toxic substances [2]. Phytoremediation uses plants’ inherent capacity to absorb, collect, and detoxify pollutants, making it a viable soil remediation method. Fresh water and healthy soil are critical resources for the survival of the environment, human life, and animals. However, as the globe becomes more technologically advanced and developed, these resources become increasingly contaminated [3]. Heavy metal pollution, especially in soil and water, is an international problem and it has a negative influence on the majority of countries [4]. Metals with an atomic mass of more than 20 are referred to as heavy metals. Examples of arsenic (As), cadmium (Cd), chromium (Cr), copper (Cu), iron (Fe), mercury (Hg), lead (Pb) and zinc (Zn) [5]. These hazardous metals’ ongoing presence in the environment increase their threat. Heavy metals can have long-term detrimental effects on both human health and the environment since they can never be totally eliminated, just transformed from one oxidation phase or organic compound to another [6]. Since the soil is the foundation of both agricultural and natural ecosystems, as well as the point where the crust of the earth and the atmosphere meet, it is vulnerable to heavy metal inputs from a diversity of sources [7].

Heavy metals in the soil are mostly derived from the dumping of industrial and urban trash, the combustion of fossil fuels, the use of untreated water for irrigation, the heavy use of chemical fertilizers, pesticides, and manures are the opposite of these [8]. More than 45 million people in Asia’s developing countries are exposed to arsenic pollution in quantities more than 50 ppb, the maximum allowed limit in drinking water in many Asian countries [9]. According to a recent research by [10], 94 to 220 million people are potentially in danger from high groundwater As concentrations, with the vast majority (94%) coming from Asia. Since India is one of the emerging Asian countries most affected by heavy metal pollution in soil, this study focuses on heavy metal removal by phytoremediation technology extensively used in India.

Maharashtra, along with two other states, is responsible for 80% of the hazardous waste produced in India, mainly heavy metal contamination, reported to the Central Contamination Control Board (CPCB) [11]. The current analysis of the scientific research on heavy metals in Indian soils from 1991 to 2018 reveals that Zn and Pb levels were higher than those recommended by the Indian government for natural soils (Zn 22.1 and Pb 13.1 mg/g) [12], While five major heavy metals—chromium, copper, nickel, lead, and iron—were discovered to be polluting Ganga, the country’s river, six other rivers—Arkavathi, Orsang, Rapti, Sabarmati, Saryu, and Vaitarna—had very high concentrations of four other pollutants [13,14]. Over 10 million sites of soil pollution have been identified globally, with heavy metals and metalloid contamination present in more than half of these areas [15]. As heavy metals build up in the environment and enter the food chain through polluted soil, water, and air, they cause ecological dysfunction and pose health concerns to both people and animals, and their impacts on plants and microorganisms are given in Table 1.

The enormous influence that environmental contaminants have on food security necessitates their repair [25]. A variety of remediation techniques have been used to lower the danger associated with heavy metal pollution so that arable land may be expanded for agricultural production to ensure the safety of the food supply [21]. The remediation strategies, which include vitrification, solidification/stabilization, phytoremediation, and microbe-assisted bioremediation are typically used to decrease the bioavailability of pollutants with heavy metals. However, various concepts underlie the repair strategies, each of which has benefits and drawbacks. These techniques’ efficacy will differ depending on the sort of soil, the type of pollutants, and the intended use of the remedied soil. Risk assessment is a potent tool that aids managers and decision-makers in managing regions that have been heavily polluted by heavy metals and contributes to the moderate preservation of our environment and health [26]. Phytoremediation is a well-known strategy for metal-contaminated soil remediation [27]. Despite their efficiency and environmental friendliness, these advances have yet to be broadly adopted in non-industrialized countries due to a lack of information or awareness. However, important concrete uses of these advances have been deemed effective and implemented in industrialized countries. The study provides a review of conventional phytoremediation, with a particular emphasis on potential improvement strategies used to increase the absorption of heavy metals utilizing phytoremediation. The generation of a list of economically viable phytoremediation species together with information on their natural range and accessibility in India is also attempted. Their prospective uses for optimizing the elimination of heavy metals from polluted places along with further financial advantages are described.

In this research, soils from both urban and rural areas are examined. Also included the tools and methods needed for phytoremediation, as well as biotechnological methods for cleaning heavy metals. Hyperaccumulator plants, or plant species having the capacity to accumulate pollutants are also briefly discussed. Examining how heavy metals affect food security, particularly the possibility for bioaccumulation and biotoxicity is a crucial component of the study. Also, a detailed summary of current advances in phytoremediation methods for removing heavy metals from damaged soil, as well as potential effects on concerns related to human and environmental health.

## 2. Status of Heavy Metals Pollution in India

Heavy metal contamination is a serious problem in India. The country is home to a number of industries that release heavy metals into the environment, including mining, smelting, and manufacturing. Also, agricultural performs the use of pesticides and fertilizers, can also contribute to heavy metal pollution [28]. All of these pesticides include heavy metals in varying proportions, which are deposited in large amounts in the soil after use [29]. Heavy metals such as Pb, Cd, Mn, and Zn are introduced into the soil by certain phosphate fertilizers. The improper disposal of wastewater for irrigation in towns and metropolitan regions with water shortages has also contributed to the contamination of agricultural lands with heavy metals. Heavy metal concentrations in soils are unacceptably high in some of India’s largest cities (Figure 1).

In India, rapid urban and industrial expansion has been observed since the turn of the 20th century. These anthropogenic activities caused a massive number of pollutants, especially heavy metals, to infiltrate surface water bodies. It enters rivers as inorganic complexes or hydrated ions that are just absorbed by silt [13]. Between 30 and 98 percent of heavy metals are found in forms that are transported into rivers by sediment [30]. There are nine primary heavy metals that damage Indian states mostly by deposition in soil; manganese and nitrate, in addition to lead and chromium, cause the greatest disruption. Furthermore, compared to the entire amount entering the soil from various sources, the amount of these metals absorbed by plants is frequently little. Zn was the element with the highest mean amount among all the vegetables examined. The heavy metals in the vegetables from these locations were in the resulting order: Cd, Cu, Pb, and Zn [29]. The accumulating metals in terrestrial systems have an impact on the health of people, animals, and plants. While heavy metals are necessary for maintaining soil health, their levels over certain permitted limits can have detrimental impacts on soil chemistry, hydrology, and biota, which have concomitant economic ramifications for soil research [31]. The previously cited heavy metals were present in the sediments in mobile and bioavailable forms that may represent a major threat by contaminating the soil and stream environment. The Indian government’s Ministry of Mines report states that the country possesses abundant resources for a wide variety of metallic and non-metallic mineral kinds. Madhya Pradesh accounted for 263 of the 1319 reported mines, with Gujarat (147), Karnataka (132), Odisha (128), Chhattisgarh (114), Telangana (39), Rajasthan (90), Tamil Nadu (88), Jharkhand (45) Maharashtra (73) and Andhra Pradesh (108) following, in order of decreasing number. In the years 2021–2022, these 11 States collectively represented 93% of all mines in the nation.

With a value of Rs. 122,142 crore, metallic minerals had a growth of around 69.18% in 2021–2022. Iron ore made up Rs 96,381 crore, or 78.91%, zinc concentrate (Rs 8182 crore, or 6.70%), chromite (Rs 4730 crore, or 3.87%), silver (Rs 4212, or 3.45%), bauxite (Rs 2477 crore, or 2.03%), lead concentrate (Rs 2237 crore, or 1.83%), manganese ore (Rs 2224 crore, or 1.82%), copper (concentrate) (Rs 1095 crore, or 0.90%), and gold (Rs 601 crore, or 0.49%) were the main metallic minerals. The output of iron ore increased by 23.86% over the prior year to 253.97 million tonnes in 2021–22. Public sector companies like as, SAIL (13.31%), NMDC (16.07%), Odisha Mining Corporation (9.01%) and others participated in about 39.30% of the overall production. 60.70% of the total was made up of the private sector, which includes Arecolor Mittal India Private Limited (2.16%), Vedanta Ltd. (2.32%), Rungta Mines (5.56%), Tata Steel (11.47%), JSW Steel Ltd. (12.45%), and others. During the year, Chhattisgarh (16.27%), Odisha (53.82%), Jharkhand (9.74%), Karnataka (15.88%), and Madhya Pradesh (2.91%) accounted for nearly all of the iron ore output (98.62%). Maharashtra, Rajasthan, and Andhra Pradesh were the stated sources of the remaining 1.38% of the production. The production of zinc concentrate climbed by 5.29% to reach 1594 thousand tonnes in 2021–2022, while the production of lead concentrate decreased by 2.36% to 368 thousand tonnes. Rajasthan was the sole producer of zinc and lead concentrates throughout this time.

## 3. Sources of Heavy Metal Accumulation

Natural elements with many atoms and an average density no less than five times higher than water are known as heavy metals. Although they may be hazardous in large doses, they are necessary for life in tiny amounts and one of the biggest environmental problems impacting people, animals, and plants is heavy metals harming water [32]. Since heavy metals are not biodegradable, they are dangerous even in low amounts [33].

Metals and metalloid ions were divided into three groups for classification. The first category contains metals that can be harmful even at small levels, such as lead, cadmium, and mercury. Arsenic, antimony, bismuth, indium, and thallium make up the second group of metals, which are less dangerous, and copper, cobalt, selenium, iron, and zinc make up the third group of metals, which are required for a variety of biochemical processes and chemical reactions in the body but only harmful above a particular level [34]. As a consequence of adsorption and, in some circumstances, inhalation, in addition to accidents or improper treatment, heavy metals build up in the soil, animal and human tissues, and other environments. According to normal biogeochemical cycles, metals were present in the world from its [35]. The existence of heavy metals in the soil sediments was caused by the underlying weathering mechanism. The soil sediments in the Maharashtra province of India are rich in zinc, cadmium, and lead because the bedrock there has highly concentrated metal deposits and mineralized veins. Stone weathering with a somewhat substantial number of heavy metals might result in metal enrichment during soil formation.

Human and manmade influences are the key sources of the raised environmental harmfulness brought on by heavy metals. Wind-borne soil particles, wildfires, eruptions of volcanoes, biogenic processes, and sea salt are a few examples of heavy metals’ natural origins [26]. According to Srivastava et al. [36], anthropogenic drivers of heavy metals pollution include mining activities, using wastewater and industrial water to irrigate agriculture fields, the usage of herbicides, pesticides and fertilizer, and the major sources of heavy metals are shown in Table 2. Fertilizers with trace amounts of heavy metals are significant contributors to these pollutants in our diet. The use of lead (Pb) as a fuel antiknock, aerosol containers, metalworking and melting, wastewater discharge, and the consumption of construction materials are examples of anthropogenic behaviors that lead to the pollution of heavy metals. Several industries, including the manufacturing of medicines, the maintenance of fiber, pulp, farming, and the creation of chloride and caustic soda, emit mercury into the environment (Figure 2) [37]. Cadmium is present in soils, rocks, coal, and mineral fertilizers to varying degrees. In the process of electroplating, cadmium (Cd) is used in a wide range of items, including textiles, batteries, metal coatings, and pigments [38]. The combined effects of these activities increase the amount of heavy metal pollution in the environment.

## 4. Heavy Metals Contaminated Impacts Environments

The contaminating effects of heavy metals are spreading worldwide. Fish can acquire heavy metals through their gills, body surfaces, or digestive systems [39]. Fish larvae and juveniles develop rather quickly, and under ideal growth conditions, enough food supply and a proper temperature are strongly associated with development in both length of body and mass [40]. On the other side, poisonous food laced with heavy metals hinders the development of fish. Growth suppression is one of the most blatant symptoms of metal poisoning in fish. Because of this, heavy metals concentrations in tissues modify numerous enzymes and metabolites, resulting in a range of metabolic, physiological, and histological alterations in fish and other freshwater animals [41]. Depending on a range of variables, including developmental agents, psychological agents, and fish lifetime, different fish species have different eating mechanisms. Fish living in contaminated environments have heavy metals accumulation in their tissues [42]. The choice of body organs for heavy metals deposition depends on a number of criteria, including metal intensity, expression duration, metal absorption, environmental factors (temperature, pH, hardness, and salinity), and intrinsic agents like fish age and eating habits. Typically, metals build up in the liver, kidneys, and gills [43]. The benthic species are killed by these poisonous sediments, which also decreases the food supply for the enormous monster. Moderate quantities of heavy metals from the environment and diet are essential for good health, while excessive amounts can be detrimental or toxic.

In addition, controlling heavy metal accumulation/concentrations in plants is a growing problem. Decoctions of medicinal plants must be prepared in a way that assures the safety and efficacy of herbal products produced throughout the process by removing harmful and non-essential heavy metals. Non-redox active metals have been found to frequently have a propensity to indirectly cause oxidative stress through a variety of ways in cascade events. These involve processes such as glutathione depletion, binding of protein sulfhydryl groups, inhibition of anti-oxidative enzymes, and even stimulation of ROS-producing enzymes like NADPH oxidases in certain plants [44]. The Haber-Weiss and Fenton processes enable the redox metals to produce oxidative damage directly. This results in the creation of ROS, or oxygen-free radical species, in plants, which damages photosynthetic pigments, disrupts cell homeostasis, breaks DNA strands, fragments proteins, and disrupts cell membrane. In the end, this may even result in cell death [45].

These heavy metals’ effects on plants result in severe growth restrictions, structural flaws, and a decline in both physiological and biochemical processes as well as plant activity. The availability of heavy metals is influenced by a number of variables, including geographic regions, element species, pH, and environmental circumstances. Not only this, but the organic makeup of the medium, fertilization, and plant species all have a direct impact on this. The metals that can be absorbed by plants are either present as soluble elements in the soil solution or have already been dissolved through root diffusion [46]. Certain metals are necessary for the growth and maintenance of plants, but too many of these metals can be poisonous. Plants can assemble the necessary metals, which allows them to obtain non-essential metals [44]. The detrimental effects of heavy metals can have an indirect impact on plant development. Because of the reduction in helpful soil microbes brought on by high concentrations of heavy metals, organic matter breakdown results in soil that is significantly less naturally fertile. Due to interference from heavy metals with the actions of soil microorganisms, beneficial enzyme activities for the metabolism of plants are also impeded. These harmful effects cause the plant to develop less quickly before dying [47].

The toxicity of plant growth and development is directly impacted by the heavy metals involved in the process. Although they do not play a very helpful function in the growth of plants, metals like, cadmium, arsenic, lead, and others do have dramatic impacts, even at very low concentrations, as has been seen in several studies. In a 2008 research, it was shown that rice plants grown in soil that contained about 1 mg/kg of mercury had significantly shorter stems. Additionally, a decrease in the tiller and pinnacle development was seen [48]. However, it was shown that cadmium toxicity resulted in a decrease in the shoot and root development in wheat plants even at soil cadmium (Cd) concentrations as low as around 5 g/L [44]. Similarly, Zou and coworkers to a thorough investigation of the effects of Cr(VI) on the cell division and development of root tips in *Amaranthus viridis* L., the mitotic index has decreased as Cr(VI) concentration has increased [44]. The incidence of c-mitosis, chromosomal bridges, anaphase bridges, and chromosome adhesion were all shown to be more common as a result of Cr(VI), which also had an impact on chromosome shape. Pea plants cultivated with Cr(VI) showed notable changes in cell cycle rates and varying levels of ploidy in the leaves. Roots displayed G2/M phase cell cycle arrest. Additionally, polyploidization was seen at both the 2C and 4C levels [49]. Studies that assessed the molecular alterations brought about by Cr(III) and Cr(VI) on the germination of kiwifruit pollen came to the conclusion that none of the Cr species had genotoxic consequences. Both caused a sharp fall in the amounts of ATP and a substantial reduction in the proteins associated with mitochondrial oxidative phosphorylation [50].

## 5. Heavy Metal Harmfulness Disturbs Human Health

The ecology naturally contains heavy metals. When present in small amounts, metals are beneficial to humans. As a result, they are referred to as vital metals. For example, in humans, Fe aids in the development of hemoglobin, Cu aids in oxygen and electron transport, Co aids in cell metabolism, Mn governs enzyme regulation, Se aids in the manufacture of hormones and antioxidants, and Ni aids in cell growth [51]. However, when heavy metal concentrations are greater, they have hazardous effects on people. Living close to a location where these metals are inappropriately disposed of, drinking water, and consuming foods polluted by heavy metals are some of the ways that heavy metals enter the human body [52]. They also enter the body by ingestion (eating or drinking) and inhalation (breathing), which have harmful consequences on people (Figure 3).

The toxicity and carcinogenic vapors of metals were said to have directly impacted a number of metal miners and on-site center employees [53]. It is well recognized that these metals are essential for enabling cells’ biological processes. One of the main processes behind the toxicity of heavy metals is the presence of reactive oxygen and nitrogen species. Damage to the liver, DNA, CNS, and kidneys occurs as a result of oxidative stress-mediated toxicity of heavy metals in both humans and animals. According to reports, heavy metals can affect the signaling cascade that causes apoptosis [54]. Specifically, A very dangerous heavy metal is cadmium (Cd). The US National Toxicology Program and the International Agency for Research on Cancer have both classified Cd as a human carcinogen [55]. Cd prevents the creation of proteins, RNA, and DNA, and it changes healthy epithelial cells into cancerous ones, making it a carcinogen. Long-term exposure to Cd in people damages the kidneys and causes lung cancer. In addition to causing functional and morphological alterations in numerous organs, interactions between Cd and critical elements including An, Fe, Ca, Mg, and Se also disrupt secondary metabolism. In severe situations, cd may result in mortality since it causes pneumonia, overall weakness, fever, and chest discomfort. Women are more impacted than males by the deposition of high concentrations of Cd in urine, blood, and kidney cortex because dietary Cd absorption in women’s intestines is higher [56].

The most often impacted systems of organs by heavy metals are the central nervous system and the digestive tract, cardiovascular, kidneys, and peripheral nervous systems. The kinds of heavy metals involved, their degree of exposure, and their chemical and state of valence can all affect the nature and severity of toxicity. Along with this, another crucial factor is the individual age as well as the method of exposure to the poison. Children’s growing neurological systems are susceptible to lead poisoning and other heavy metal toxicity [57]. Similarly, the effects of nickel (Ni) are detrimental to human health. Ni mostly causes lung cancer, chronic bronchitis, and lowers lung function. Nickel oxide, a carcinogenic form of nickel, causes lung cancer, asthma, and sinus issues when it is inhaled over an extended period of time [58]. The kidney is the organ that is most impacted by nickel exposure. Acute nickel poisoning results in Frank haematuria, kidney damage, alveolar cell hyperplasia, and congestion in the lumen, as well as carcinogenesis, chromosomal damage, mutation, and inhibition of NK cell function as Ni^2+^ serves as a tumor [59]. Ni damages human lungs by causing bleeding, edema, disorganized alveolar cells, pulmonary fibrosis degeneration, and bronchial epithelium degeneration [60]. Adult respiratory distress syndrome (ARDS) brought on by nickel has occasionally resulted in death.

Heavy metal arsenic is poisonous to humans. Carcinogen status is known for inorganic arsenic (As). Diabetes, hepatic and renal failure, and neurological issues are brought on by low to moderate amounts of As exposure. Women are more prone than males to As-induced skin diseases because their skin is thought to be more vulnerable to the substance, which produces dermatitis [45]. Keratosis, melanosis, and pigmentation are skin lesions that are indicative of As exposure. Furthermore, Mercury (Hg) is the most dangerous non-essential metal for humans. In the environment, Hg exists in three forms: elemental Hg, inorganic Hg, and organic Hg. Elemental Hg is mostly emitted as a vapor in the environment [61]. Elemental Hg vapor primarily affects the central nervous system, causing cognitive, motor, and sensory disturbances as well as symptoms such as sleeplessness, memory loss, tremors (affecting hands and occasionally other areas of the body), and muscle twitching. Extended contact with elemental Hg results in impaired focus, clouded eyesight, and shaky walking.

Several researchers have noted that exposure to excessive quantities of heavy metals may even go so far as to increase the generation of free radicals, resulting in oxidative stress. The production of oxidative stress is one of the main processes behind the toxicity of heavy metals [62]. Humans can directly consume heavy metals through contaminated food, marine life, contaminated water, breathing in dust emissions, or exposure while at work. Heavy metal contamination chains nearly always follow a cycle, going from industrial to the atmosphere, land, water, and food, then to people.

## 6. Techniques for Detecting Heavy Metals

A heavy metal ion detection is an equipment or gadget designed to detect metal ions in its vicinity. Metal ions can sometimes be measured using a metal ion detector method. Before eliminating the harmful ions contained in water samples, a technology for detecting the amount of metal ions must be provided. The strategies should also help in the quantitative estimate of the pollution level, allowing for the simultaneous identification of an acceptable removal procedure. In this regard, a detection procedure that is both time and cost-efficient as well as ecologically safe should be devised. To identify metal ions traces, a detection procedure must be highly sensitive and precise. There are several ways to detect heavy metal ions: however, none of them detect all of the ions present in the sample. The following section discusses several heavy metal detection techniques.

Inductively coupled plasma mass spectrometry: ICP-MS is a very sensitive and precise technology for identifying a wide spectrum of heavy metals in a wide range of matrices. ICP-MS analyzes a sample by ionizing it and then detecting the ratio of mass to charge of the ions [63]. For ICP-MS is costly and needs specialist equipment, it is primarily employed in laboratories.Atomic absorption spectroscopy (AAS): Another sensitive and precise approach for detecting heavy metals is AAS. AAS works by absorbing certain wavelengths of light into the sample. The percentage of light absorbed is related to the metal content in the sample. AAS is also commonly utilized in laboratories. Researchers have conducted a more thorough study on AFS as a result of the continued development of this technology. At the moment, these disciplines include food, medicine, agriculture, health, and the prevention of epidemics, as well as the environment [64].X-ray fluorescence (XRF): A non-destructive method for finding heavy metals in a wide range of matrices is XRF. X-rays are used to irradiate the sample in XRF, which subsequently measures the fluorescence energy the sample emits. Since it is very portable and does not need substantial sample preparation, XRF is frequently utilized for on-site analysis. Heavy metals in solid materials, such as soil, minerals, and relics from ancient civilizations, are analyzed using XRF [65].Assays using colorimetry: Assays using colorimetry are straightforward and reasonably priced methods for identifying heavy metals. A reagent is added to the sample in a colorimetric test, which changes color when the metal is present [66]. Since colorimetric tests are portable and do not need specific equipment, they are frequently utilized for field testing. These techniques are used to quickly and simply analyze the presence of heavy metals in biological, soil, and water samples.Biosensors: Biosensors work on the premise that biological molecules like enzymes and antibodies can detect heavy metals. Biosensors are frequently very sensitive and selective, and they may identify heavy metals in complicated matrices like blood and urine [67].

The approach used to detect heavy metals will be determined by a variety of parameters, like the kind of material to be examined, the required sensitivities and precision, and the budget that is accessible.

## 7. Phytoremediation for the Removal of Heavy Metals from Contaminated Soil

Phytoremediation is regarded as an efficient, aesthetically pleasing, cost-effective, and ecologically sound approach for heavy metal cleanup in the environment. Plants that use phytoremediation gather pollutants in their roots and transport them to the body surface [68]. In order to encourage plant growth and cleanse soil and water, phytoremediation mixes naturally occurring or genetically modified plants with the matching rhizospheric bacteria. Through chelating these pollutants inactively in the soil or linking them in their tissues, plants normally deposit these pollutants in vesicles away from the delicate cell cytoplasm where many metabolic processes take place [69]. Plants are used in phytoremediation to phytoextraction, phytostabilization, and phytovolatilization of heavy metals [70]. The three main techniques used for phytoremediation are as follows: (i) Phytoextraction is a widely used technique that uses accumulators’ heavy metals acceptance ability to allocate heavy metals from soils to plant parts; (ii) Phytovolatilization of heavy metals from plant tissue to atmosphere occurs when heavy metal accumulated is highly volatile and (iii) Phytostabilization approach reduces heavy metals movement inside [71]. Figure 4 depicts the removal mechanism of heavy metals through the phytoremediation process in plants. These are divided into three kinds, including excluders, accumulators, and indicators, depending on how they respond to higher HM levels [72]. The accumulator aids in the translocation and absorption of HMs to above-ground plant parts, signals manage the process of bioaccumulation of HMs to show internal content, and excluders restrict the absorption and transfer of HMs in roots to provide a phytostabilizing effect.

Plants ideal for phytoremediation may extract large amounts of heavy metal contaminants into their roots, allowing them to survive different metals and adapt to different conditions [27]. Eid et al. [73] investigated the ability of various aquatic hydrophytes, *P. australis, E. crassipes, L. stolonifera*, and *E. stagnina*, to accumulate Cd, Ni, and Pb and use these plants for signaling and phytoremediation of heavy metal-contaminated wetlands. The results showed that Phragmites australis collects the most Cd and Ni, although Echi-nochloa stagnina accumulates the most Pb in the tissues. In addition, these researched species are capable of phytostabilizing these tested heavy metals, with the exception of Ni in *Echinochloa stagnina* and Cd in *Ludwigia stolonifera*, which are suited for phytoextraction of these metals. Consequently, phytoremediation is a practical and affordable method for removing HMs from soil and water. Particularly in India, the heavy metal accumulation ability of different terrestrial species found species growing dominantly at highly polluted sites have high metal accumulation in comparison to rare and common status species. Amongst the most dominating terrestrial plants *Cannabis*, *Croton*, *Sachhrum*, *Cassia*, *Parthenium*, and *Eliptica* were found most efficient heavy metal tolerant plant [74]. Metal accumulation by five aquatic plants growing naturally in water bodies in the FA vicinity was also observed. In aquatic plants, metal accumulation was maximum in *Typha latifolia* as compared to the other four species, i.e., *Azolla pinnata*, *Hydrilla verticillata ceratopteris thallicteroides,* and *Marsilea minuta*.

### 7.1. Genetic Modification in Plants

Transferring the genes involved in metal absorption, transport, and sequestration into suitable plants is one method for genetically modifying plants to have better phytoremediation features. Transgenic plants that are designed to accumulate high quantities of metals in obtainable portions may be created, depending on the technique. Gene transfer or overexpression will result in improved intracellular targeting, sequestration, translocation, and absorption of metals. Plants that have been genetically engineered to produce metal chelators will be better able to absorb metals [75]. Genes from hyperaccumulators or other sources can be transplanted to generate effective transgenic plants for phytoremediation. The remediation technology has been thorough recently and provides suitable methods for cleansing contaminated soils [76]. The transgenic approach is limited to transforming functional genes and selecting specific promoters, potentially enhancing the gene functions linked to heavy metal accumulation, transfer, or detoxification processes. Typically, plants absorb toxic substances through a variety of mediums, including water, soil, and air [77]. Heavy metal accumulation is a complicated process with several stages that allow the metals to enter the root cells through a membrane transporter protein. According to Sharma et al. [78], heavy metals are carried into the xylem of plants for purification and compartmentalization into appropriate aerial sections. The greatest markers of this intricate process are the heavy metals’ bioavailability and translocation to the root-effective absorbing region. Metals in the soil in reactive ion form are the first step in the process of metal accumulation in plants, which can have beneficial or detrimental effects.

Regardless of whether metal ions are beneficial or dangerous, proton (H^+^) excreting in the milieu of root rhizospheres (amino acids, enzymes, and phytochelatins) increases the bioavailability or mobility of metal ions [79]. In the real root sorbing zone, the heavy or other beneficial ions move via the mass flow route of the soil liquid phase, which is regulated by transpiration and the pace of ion diffusion in the root-surface cell. Metal ion entrance into ground root cells is catalyzed by membrane transporter proteins. Because they mimic the necessary tiny components and the poorly target-specific membrane proteins, excessive concentrations of unnecessary substances in soils can seep into plants [80]. Metal-binding agents, such as metallothioneins and phytochelatins, which include organic acids and amino acids chelation; including the activation of enzyme production for reactive oxygen species, glutathione and phosphate-derived production growth over cell wall binding and compartments are some of the ways that metal/metalloids detoxification is mediated. Stressful conditions cause the structure of plant cell walls to remodel to facilitate lignification and serve as a storage reservoir for heavy metals [81].

In terms of phytoremediation, biotechnological technologies such as genetic modification and molecular science offer enormous potential to change or alter microorganisms and plants. Numerous researches on the genetic alteration of plants for the phytoremediation against various contaminants have been carried out with success. Genes linked to the relationships of microbes, the resistance or degradation of contaminants, and the increased biomass of plant hosts can be regulated or overexpressed by genetic engineering. It has been demonstrated that by overexpressing glutamylcysteine synthase, genetically engineered plants such as *P. angustifolia*, *N. tabacum*, and *S. cucubalis* collect more heavy metal pollution than their wild counterparts [82].

### 7.2. Phytoextraction

The process known as phytoextraction, sometimes known as phytoaccumulation, entails the uptake of hazardous metals by plant roots, which are then transferred to shoots and deposited at vacuoles, cell walls, cell membranes, and other areas of the plant tissues that are not metabolically active. Known for their ability to remediate heavy metal accumulation, hyperaccumulator plants store more harmful metals in the tissues of their roots and shoots. The uptake of metal cations is one of the general pathways during the buildup of toxic heavy metals (HMs). This is followed by the creation of metal-ligand complexes or metal-phytochelatin complexes (M-PC) inside plant cells [83]. These complicated molecules get transferred to the plant vacuole and kept thereafter generating the M-PC. Plant biomass and the amount of heavy metals (HMs) in above-ground plant tissues are the primary determinants of a species’ capability for extraction [84].

As a result, the ideal species for phytoremediation must not only be able to withstand and efficiently absorb HMs, but also develop quickly, produce a large amount of biomass, and offer financial advantages [85]. Thus, for the purpose of phytoremediation of certain metals, some plant species are better adapted to use the phytoextraction method (Table 1). Examples of plants that have been shown to use phytoextraction principally for many metals include *Commoelina communis* for Cu, *Pteris vittata* for As, *Sesbania drummondii* for Pb, *Sedum alfredii* for Cd, etc. It has been discovered that some bacteria that encourage plant development, such as *Agrobacterium* sp. and *Stenotrophomonas maltophilia*, increase arsenic (As) phytoaccumulation in *Arundo donax* [86].

### 7.3. Endophytic Microorganisms Utilization

Interactions among soil and plant microbes in phytoremediation have good benefits since it is a low-cost strategy with a minimal risk of environmental impact [87]. Endophytes are microorganisms that inhabit the interior tissues and intercellular gaps of plants without producing disease [88]. Herbaceous plants are widely used to separate endophytes. Idris et al. [89] presented the first investigation on the separation of endophytic bacteria resistant to heavy metals. The scientists, including Halácsy, extracted endophytic microbes from the interior of *Thlaspi goesingense*, a nickel hyperaccumulator. The research was carried out in eastern Austria, where the overall nickel level per kg of soil was 2.5 mg. The isolates were divided into two groups: *alphaproteobacteria* and Gram-positive bacteria. Approximately 42% of the isolates had a high degree of similarity with the *Methylobacterium mesophilicum* species and 37% with *Sphingomonas* sp. Other isolates shared similarities with *Rhodococcus*, *Curtobacterium*, and *Plantibacter*. El-Deeb et al. [90] identified endophytic microbes of the Enterobacter genus from the Egyptian aquatic plant *Eichhornia crassipes*.

Resistance to zinc, cadmium, and lead was found in the bacterial strains. Sun et al. [91] identified endophytic bacteria from rapeseed (*Brassica napus*) growing in the Nanjing suburbs in 2008. The soil from where the plants were gathered included the greatest concentrations of lead (216.5 mg/kg) and zinc (204.5 mg/kg). Lead-resistant bacteria were isolated from rapeseed, with the major strains being *Pseudomonas fluorescens* and *Microbacterium* sp. The bacteria also aided plant development by producing plant hormones, dissolving lead, producing siderophores, and producing 1-aminocyclopropane-1-carboxylic acid deaminase [91]. Furthermore, certain endophytic bacteria that dwell on the root surface aid in *rhizofiltration*. Inside the plant root surface, certain bacteria of *Ochrobacterium* and *Pseudomonas* converted hexavalent Cr (Cr-VI) to trivalent Cr (Cr-III). The decreased Cr-III is subsequently easily precipitated inside the plant root, effectively cleansing the water [92]. The bioconcentration factor for many metals by aquatic plants appropriate for rhizofiltration applications reached as high as 0.001948 of the plant’s dry weight. By releasing organic ligands, digesting organic materials, and emitting metabolites and siderophores, soil microorganisms can improve metal solubility and oxidation [93].

According to Abou-Shanab et al. [94], the presence of a particular microbiota boosted the phytoextraction of nickel by Alyssum murale. Microorganism-produced low-molecular-weight organic acids, such as citrate, succinate, acetate, 2-ketoglutarate gluconic acid, oxalate, malate and play an important role in heavy metals mobilization. Whiting et al. discovered that inoculating soil by metal-resistant rhizosphere bacteria greatly boosted zinc ion availability and accumulation in plants. Siderophores, which are low-molecular-weight organic chelators with a high affinity for iron ions Fe^3+^ and are generated by microbes in an environment of iron Fe^2+^ deficiency, play a significant role in metal mobility. Metals linked by bacterial siderophores can be absorbed by bacteria and plants, increasing metal buildup in plant tissues. Pyoverdine, for example, is produced by microorganisms of the *Pseudomonas* species.

## 8. Recent Biotechnological Approaches for Potential Phytoremediation of Soil

Plants are being used for hazardous metal cleanup; however, as a result of HM phytotoxicity, this method of cleanup has been slow and typically failed [95]. Phytoremediation, which uses genetic engineering approaches to enhance plant tolerance and toxic metal accumulation, has a lot of potential. Furthermore, a host of new research has been conducted using omics technology to identify the genetic components and underlying processes in plant heavy metals tolerance. Hg, Cd, Cu, As, Se, and Pb have been phytoremediated utilizing biotechnological approaches. Using three basic biotechnological approaches, plants are being modified for the phytoremediation of HMs. These strategies include modifying heavy metals transporters genes and uptake systems, improving HM ligand production, and changing heavy metals into a reduced amount of toxic and unstable systems.

Improved tolerance and heavy metal accumulation in numerous plant species have been brought about by modifications to several heavy metal transporters. Over-expression of the YCF1 gene in *Arabidopsis thaliana* increases plant heavy metals accumulation and improves resistance to Pb and Cd [96]. NtCBP4 protein from Nicotiana tabacum was overexpressed in transgenic plants, improving Pb hypersensitivity and accumulation. Additionally, increased production of an NtCBP4 mutant with a shortened amino acid sequence enhanced Pb tolerance but reduced its accumulation [97]. Additionally, T-DNA mutants of the *Arabidopsis CNGC1 gene*, which generates a protein related to NtCBP4, demonstrated Pb tolerance. These results show that Pb^2+^ transport involves NtCBP4 and AtCNGC1 [98]. Higher Mn, Cd, and Ca ion accretion was the outcome of *Nicotiana tabacum* expressing the CAX2 gene. In isolated root tonoplast vesicles, CAX2 overexpression improved the transport of Cd and Mn ions as well [99]. The transcription of numerous and distinctive genes was elevated in rice roots exposed to HM such as As(V), Cd, Cr(VI), and Pb [100]. A de novo transcriptome sequencing technique was also used to identify Cu-resistant genes in the *Paeonia ostii* plant. Similarly, *D. viscosa* (L.) *Greuter* was found to possess a *DvNip1* gene homolog. The expression of *DvNip1.1* in *D. viscosa* clonal lines cultivated under different stressful conditions was also observed, and the proportion of *DvNip1.1* translation levels found in both shoots and roots is suggested as a plausible selective marker for recognizing highly resistant *D. viscosa* plants [101]. Previously Anglana et al. [102] investigated *D. viscosa*’s ability to absorb, translocate, and proliferate in different levels of Cd^2+^, As^3+,^ and As^5+^. In fact, for As and Cd, we found opposing patterns of phytoextraction from the aerial portion and biological concentration in the roots.

Plant species can also be modified to boost resistance to As, Cd, and Pb. Three times more transgenic crops were gathered. When two bacterial genes were co-expressed in Arabidopsis and cultivated in a medium containing sodium arsenate (125 M), the results were similar to those seen in above-ground biomass [103]. There is a lot of promise for improving heavy metals and metalloid resistance by altering genes to boost the production of metal chelation agents [104]. In *A. thaliana* and *B. juncea*, the constitutive expression of AtPCS1 boosted arsenate tolerance but did not increase shoot growth. due to accumulation [105]. Despite being a necessary element, selenium (Se) is harmful in greater amounts and is thus regarded as a worldwide contaminant [106]. Dimethyl selenide (DMSe), for example, is created by plants from selenium (Se). Se tolerance, volatilization, and bioaccumulation have been improved using biotechnological Se phytoremediation. Recently, Metal Binding proteins (MBPs) technology getting attention in India and plants possess several metal-binding proteins (MBPs) that aid in the absorption, build-up, transfer, and detoxification of heavy metals, hence enabling the plants to tolerate these harmful metals [107].

Overall, using these tools and methods can improve phytoremediation efficiency while rendering it a more attractive alternative for cleaning up polluted lands. To determine its efficacy and safety for a specific location, phytoremediation must be studied case-by-case, just like every other remediation approach. Before executing a phytoremediation method, factors including soil properties, pollutant kind and climate, concentration, and species of plants should all be considered.

## 9. Factors Impacting Heavy Metals Phytoremediation

Plant species, root position, weather, element species, and soil physiochemical and biochemical variables all influence HM synthesis and distribution in plants. To improve cleaning, agronomic measures have been implemented (pH changes, chelators, and fertilizers are a few options). Additional crucial parameters to consider are the soil pH, organic matter, and phosphorus level. By adding lime to the soil, the pH may be raised to 6.5–7.0, which will reduce the amount of Pb that plants absorb [108]. Moreover, as will be explained below, a variety of variables affect a plant’s capacity to absorb HMs.

## 10. Conclusions

Heavy metal pollution can have detrimental effects on the environment, leading to quick deposition and posing a serious threat to agricultural output and food safety. Since heavy metals are vital micronutrients for plants but hazardous at larger levels, it is crucial to establish minimum and maximum heavy metals standards for water used for irrigation and agricultural land. There are several strategies that have been developed to reduce heavy metal contamination and replant damaged soil. In many developed countries, phytoremediation is viewed as a viable strategy for replanting soil that has been contaminated by heavy metals due to its high degree of public acceptability and several advantages over alternative physicochemical treatments. Furthermore, a few of the opportunities and strategies for improving phytoremediation, such as different phytoremediation methods, biotechnology techniques, plants used in different approaches, and variables influencing phytoremediation, have been discussed, opening up possibilities for the establishment of new strategies. Phytoremediation remains in existence in the developing country of India. The increase in research in this area is anticipated to aid in the cleanup of hazardous places and benefit the economy and more study is required to better understand the effects of various kinds of catalysts on phytoremediation efficacy in order to boost phytoremediation practicability for environmental restoration.

## Figures and Tables

**Figure 1 plants-12-03816-f001:**
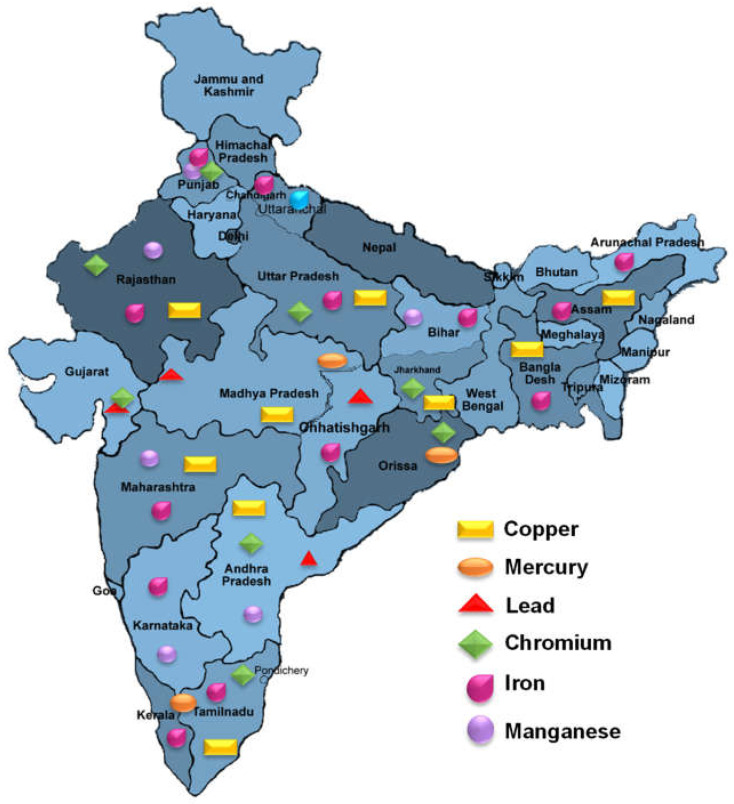
State-wise impacts of heavy metals in India.

**Figure 2 plants-12-03816-f002:**
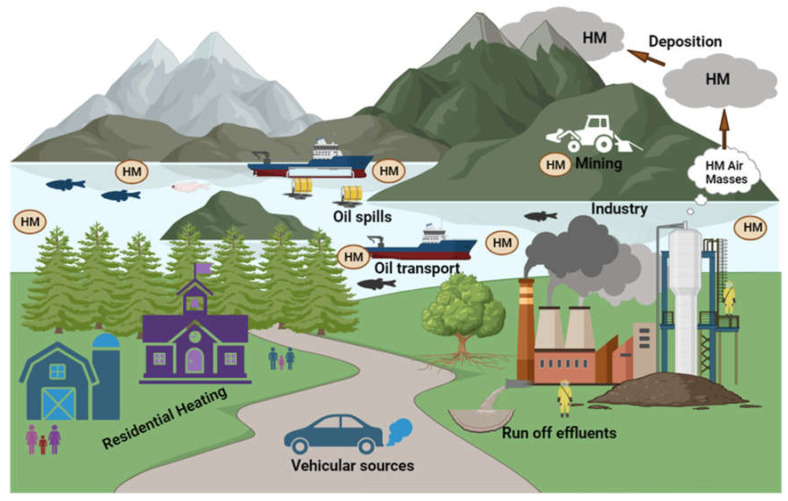
Heavy metals accumulation from different sources.

**Figure 3 plants-12-03816-f003:**
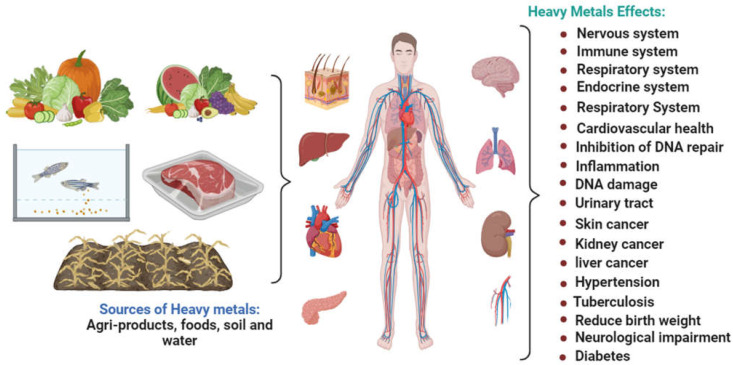
Health impacts with toxic exposure to heavy metals from contaminated natural sources.

**Figure 4 plants-12-03816-f004:**
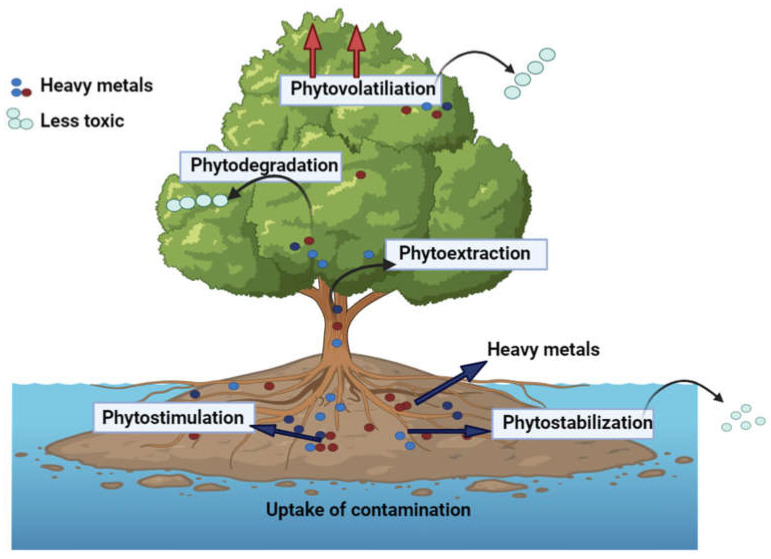
Mechanism of phytoremediation for heavy metal removal.

**Table 1 plants-12-03816-t001:** Impact of heavy metals on microorganisms and plants.

Heavy Metals	Impacts of Heavy Metal on Microorganisms and Plants	References
**Arsenic**	Arsenic causes the deactivation of enzymes in bacteria. Reduced shoot and root development, necrosis, chlorosis, senescence of the leaves, defoliation, limited stomatal conductance and nutrient absorption, and degradation of chlorophyll	[16]
**Copper**	Disrupts cellular function and restricts the actions of enzymes in microorganisms. Cu affects oxidative stress, chlorosis, and hinders plant development.	[17]
**Cadmium**	Nucleic acid is impacted, as are the division of cells and transcription, the mineralization of carbon and nitrogen, and the denaturation of proteins in microorganisms. A higher toxicity stops plants from growing and causes plant necrosis. Exposure to Cd in soil produces an osmotic stress response in plants, which damages their physiological health by lowering transpiration, stomatal conductance, and leaf relative water content.	[18]
**Chromium**	Reduction of Cr(VI) to Cr(III), Biosorption, precipitation, reduced accumulation and chromate efflux are only a few of Cr-resistance mechanisms that microorganisms and likely plants exhibit. Growth, an extension of the lag phase, and a decrease in oxygen intake.	[19]
**Lead**	Protein and nucleic acid degradation interfere with transcription and enzyme activity. Fruits and vegetables cultivated in soils polluted with high amounts of lead may be the source of lead poisoning.	[20]
**Mercury**	Population size reduction, protein denaturation, cell membrane instability, and the function of enzymes in microorganisms. Reduces plant development, persuades genotoxic effects, increases lipid peroxidation, yield, nutrient absorption, homeostasis, and oxidative stress.	[21]
**Nickel**	Interferes with the actions of enzymes in microorganisms and disturbs cell membrane. a study by E. coli, many bacteria are worried by ambient nickel in usual conditions. Decreases growth and nutrient absorption, enzyme activity, and chlorophyll content.	[22,23]
**Zinc**	Execution, obstruction to biomass, and microbial development Plant biomass, chlorophyll content, growth rate, germination rate, reduces photosynthesis	[24]

**Table 2 plants-12-03816-t002:** List of toxic Heavy metals and their production sources.

Heavy Metals	Sources
Arsenic (As)	Natural processes/Geogenic, fuel burning, thermal power plants, smelting operations
Chromium (Cr)	leather tanning, chromium salts manufacturing, industrial coolants, Mining
Copper (Cu)	Smelting operations, Mining, electroplating
Fluoride (F)	Water additive, industrial waste, Natural geological sources
Lead (Pb)	E-waste, smelting operations, paints, Lead acid batteries, coal- based thermal power plants, bangle industry, ceramics,
Mercury (Hg)	Thermal power plants, Chlor—alkali plants, electrical appliances fluorescent lamps, hospital waste (Sphygmomanometers, barometers damaged, thermometers)

## Data Availability

Not applicable.

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
