# Peer review of "Current Status of Biotechnological Approaches to Enhance the Phytoremediation of Heavy Metals in India—A Review"

_plants, 2023, doi:10.3390/plants12223816_

Round 1

Reviewer 1 Report

Comments and Suggestions for Authors

13-22 Re-manage because it isn’t easy to understand and use more appropriate world, like “contribute” instead of “subsidize”, or “environment” instead of “settings”. Sometimes the sentence aren’t very well connected.

20 avoid “forms of”: hyperaccumulator plants are a well known  group of plants.

26 "extensively" instead of "existence".

27 “as heavy metal cleansing” is superfluous and make confusion in the sentence

28 “in developing country of India”. ”country “ is correct, but could be more appropriate use “land” or “region”. You can also cite the land/region involved.

29 what you means with “bids”?

35 “Contaminants” instead of “toxin”.

37-38 The sentence is incomplete and repeat what you write before.

38 “cause” instead of “offers”.

43-45 Some references or rearrange; in 55-58 you explain better what probably you wrote here.

47 Remember that As is a metalloid

49 The sentence repeat what you will write better in the following sentences.

57 What do you mean for “garbage from automobiles and garages”? Exhaust gas from automobiles?

58 The opposite of what?

58-60  Repetition of what you wrote before.

73 Probably is mg/g not g/g.

93 “health” instead of “overall wellbeing”.

100 “creation” or ”generation” instead of “compilation”.

104-118 A lot of repetition regarding the aim of the review.

145 “previously cited” instead “aforementioned”.

Figure 3. Some image on the left are stretched.

Lack of references 297-306, 325-331, 355-372.

In general, the aim of the article it isn't clear; Do you want to focus on phytoremediation in the specific case of India or not? In both case the aim it isn't achived. In the Chapter 7, you miss to cite a lot of other plants and the unique that you cite come from a stady not related to the case of India.

Write a review it isn't easy and need a lot of effort. You did a good bibliograpy reserch but it isn't sufficent; expand and focus better on the India case of study with more approprite references.

Very often you repeat same things only changing the sentences.       

Comments on the Quality of English Language

There are a lot of unusual world and sometimes is very difficult to follow the story.

Reviewer 2 Report

Comments and Suggestions for Authors

The authors need to proofread the manuscript thoroughly. I suggest the following changes and improvements:

1.     Line 58-60: Modern society's heavy metal pollution of the water and soil environment has been accelerated by industrialization, the world's fast population rise, and increased agricultural……. Please rephrase and eliminate the grammatical issue.

2.     The novelty and research significance of this review should be stated clearly in the introduction section.

3.     Add a section regarding the literature review and search results. Please clarify that from the ### articles you found up to this point, you made a first assessment by screening titles and abstracts. When was this research conducted? This is very important because databases are updated also for past periods.

4.     Line 188: Authors need to improve the legend of Table 2.

5.     Heavy metal pollution is a harmful effects and quick deposition in the environment……… Please rewrite and eliminate the grammatical issue.

6.     A brief Table should be added to each section summarizing the key results.

7.     The authors need to provide detailed research direction advising young researchers, especially Ph.D. students who would be reading with interest to learn more about this research topic.

Comments on the Quality of English Language

I have mentioned it in the comments section.

Reviewer 3 Report

Comments and Suggestions for Authors

The Review article “ Current status of biotechnological approaches to enhance the phytoremediation of Heavy metals- a review” investigates phytoremediation techniques for in situ and ex situ treatment of heavy metal contaminated soil. The figures and tables support the publication, however, the article needs improvements. I have some comments that can be helpful for the revision.

For example.

Title. The article is about approaches to increase phytoremediation of heavy metals in india, the introduction is all about indian references, but the title lacks it. So the country name INDIA should be included in the title, else the review should present a global scenario, which is not the case here.

Abstract.  Line 13-28 is all about introduction or background, only 2-3 lines are enough, what the reader wants to see here, what the review discusses, but the objectives are only presented in last 2 lines. So explain about what this review is all about, not the introduction in the abstract.

Introduction. If the review is about a global scenario. why the examples are quoted about INDIA only, line 68-78 is an example. Keep it global or keep it country centric  

A table in the introduction section…it is quite general, just remove it, as most of the impacts presented in that are repeated in the article and the impact of each heavy metal is similar in many aspects., if you have to include this, just summarize in a small paragraph in very few lines.

2. status of heavy metals in india. The figure is very general and the prospect provided here too. What the reader is interested to see is how much concentration of these metals in each area of the country ? the authors need to analyse the data to present some kind of meta analysis at least that how many case studies have been conducted previously and what is the current situation ? the concentration ? reasons ? for that in the particular area ? major aspects ?

3. source of heavy metals. The section is very general once again. The readers know about it already. This should have been summarized in intro in a small paragraph,

Intro could start with what are heavy metals, what are sources in next paragraph, then how the phytoremediation measures can be conducted etc

Part 4, 5, 6 is also just introduction. Its going on and on and on, and the main theme is phytoremediation which comes at NO.7 ….that actually deserves the main attention as per your title.

But only one page is written about it. Surprisingly. And also there is no approaches mentioned clearly as 1, 2, 3, 4 etc etc. For example,

1.Genetic Modification in plants:

2. Endophytic Microorganisms utilization:

3. Improving Pollutant Transformation in soil by Phytoextraction Enhancement and Rhizosphere Engineering

Nothing as I mentioned here has been added about. These sections must be added each containing atleast 1000 words with figures, if this article needed to be published.

Minor comments

Typo errors. The article is full of such errors, for example, see line 48. Please revise the article carefully.

Gene names should be italic

Comments on the Quality of English Language

OK

Round 2

Reviewer 1 Report

Comments and Suggestions for Authors

I saw some improvements and appreciate that you took my advice. Unfortunately, there is a lot of repetition of the same theme steal as well as unusual construction such as "garbage from automobiles" (line 55). Figure 3 is also still poorly arranged (the fruit and other images are visibly streatched). You may also include the studies on NIP1;1 in Dittrichia viscosa (Anglana et al 2023 10.3390/plants12132499 and De Paolis 2022 10.3390/plants11151968) in Chapter 8. Another plant that could be used for phytoremediation is Dittrichia Viscosa (Papadia et al., 2020 10.1080/11263504.2020.1836061).

In the conclusion tray to more closely link the heavy metal pollution in India with the findings of your research and provide advice on which techniques and plants to employ in the particular instance of India, taking into account the growing conditions such as soil, climate, etc.

Comments on the Quality of English Language

The English structure has improved over the previous version, but it still needs to be enhanced more.

Reviewer 3 Report

Comments and Suggestions for Authors

article is improved after the revision, but plagiarized 25%

Comments on the Quality of English Language

ok

Round 3

Reviewer 1 Report

Comments and Suggestions for Authors

Dear authors,
now the article is more complete and focused on the topic. It will be an excellent starting point for future articles regarding heavy metal pollution. I saw that you followed my advice and I appreciate it. Last things, don't you have institutional emails to indicate in your affiliations? Wouldn't it be better to transform raw material prices into dollars/euros instead of Indian rupees?